# Adapting cognitive diagnosis computerized adaptive testing item selection rules to traditional item response theory

**Miguel A. Sorrel**[1], **Juan R. Barrada**[2]*, **Jimmy de la Torre**[3], **Francisco José Abad**[1]

**1** Department of Social Psychology and Methodology, Universidad Autónoma de Madrid, Spain, **2** Department of Psychology and Sociology, Universidad de Zaragoza, Spain, **3** Faculty of Education, The University of Hong Kong, Hong Kong

* barrada@unizar.es

**Data Availability Statement:** The open database and code files are available at the Open Science Framework repository (https://osf.io/kagxy/).

## Abstract

Currently, there are two predominant approaches in adaptive testing. One, referred to as cognitive diagnosis computerized adaptive testing (CD-CAT), is based on cognitive diagnosis models, and the other, the traditional CAT, is based on item response theory. The present study evaluates the performance of two item selection rules (ISRs) originally developed in the CD-CAT framework, the double Kullback-Leibler information (DKL) and the generalized deterministic inputs, noisy "and" gate model discrimination index (GDI), in the context of traditional CAT. The accuracy and test security associated with these two ISRs are compared to those of the point Fisher information and weighted KL using a simulation study. The impact of the trait level estimation method is also investigated. The results show that the new ISRs, particularly DKL, could be used to improve the accuracy of CAT. Better accuracy for DKL is achieved at the expense of higher item overlap rate. Differences among the item selection rules become smaller as the test gets longer. The two CD-CAT ISRs select different types of items: items with the highest possible $a$ parameter with DKL, and items with the lowest possible $c$ parameter with GDI. Regarding the trait level estimator, expected a posteriori method is generally better in the first stages of the CAT, and converges with the maximum likelihood method when a medium to large number of items are involved. The use of DKL can be recommended in low-stakes settings where test security is less of a concern.

## Introduction

Computerized adaptive testing (CAT) is one of the applications of the item response theory (IRT) that has received greatest attention in the recent and past literature (e.g., [1–3]). A CAT consists of a specifically tailored set of items in which each of the items is selected to be administered on the basis of the responses of the examinee to the previously administered items. Thus, each examinee may have a different set of items, those that are most informative with respect to their ability estimates. The main advantages of CAT include a reduction in testing

**Funding:** This research was supported by Grant PSI2017-85022-P (Ministerio de Ciencia, Innovación y Universidades, Spain) and the UAM-IIC Chair «Psychometric Models and Applications». There was no additional external funding received for this study.

**Competing interests:** The authors have declared that no competing interests exist.

time, an improvement in the accuracy with the same number of items compared to a fixed test, and an increase in test security. Early CAT applications were based on unidimensional-IRT models for dichotomous data.

As a result of the evolution of the psychometric theory, the emergence of new models has been made possible the application of CATs with different response formats (e.g., polytomous, continuous, forced-choice), and tests assessing more than one dimension using multidimensional-IRT and bi-factor modelling. This has enabled the development of CATs based on these models. Thus, we have, for example, the Tailored Adaptive Personality Assessment System [4], a multidimensional forced-choice CAT for evaluating the Big Five personality traits and some of their facets. Another example is the CAT Depression Inventory [5] which is based on the bi-factor model.

What all the above-mentioned developments have in common is that the underlying latent traits are assumed to be continuous. Nonetheless, adaptive methodologies have been recently applied to a new psychometric framework: the cognitive diagnosis modeling (CDM) framework. This new set of models emerged with the purpose of diagnostically classifying the examinees into a predetermined set of discrete latent traits, typically denoted as attributes. Attributes are discrete in nature rather than continuous, with usually only two levels indicating if the examinees mastered or did not master each specific attribute. CDM is a very active area of research (e.g., [6,7]). Some of the latest developments are in the area of cognitive diagnostic CAT (CD-CAT). However, compared to the large amount of research in the traditional IRT context, to date only a small number of research has been conducted in the context of CD-CAT (e.g., [8–12]).

The Fisher information statistic is the most widely used method for item selection in traditional CAT [13]. This method requires continuous ability levels. Because attributes in CDM are discrete, Fisher information cannot be used in a CD-CAT setting. Fortunately, there exist alternative methods that can deal with that. One of these methods is the Kullback–Leibler (KL) information. Different modifications of KL item selection rule (ISR) have been proven to be useful in the CD-CAT context (e.g., [10]). Furthermore, new ISRs such as the global-discrimination index (GDI; [10]) have been developed. The present study uses Monte Carlo methods to evaluate the performance of these rules generated in the CD-CAT within the traditional IRT framework.

The remainder of the manuscript is organized as follows. First is a review of the ISRs that are used in the present study. This is followed by a presentation of the simulation study designed to illustrate the performance of these ISR. Finally, the results of the simulation study are discussed, and several research limitations and possible future directions are provided.

## Item selection rules

**Point Fisher Information.** Several ISRs have been proposed in the traditional IRT framework. Among them, point Fisher Information (PFI) is the most popular one. This method consists in maximizing Fisher information at the current estimate of the latent trait level (i.e., $\hat{\theta}$; [14]). Specifically,

$$j = \arg\max_{i \in B_q} I_i(\hat{\theta}), \tag{1}$$

where $I_i(\hat{\theta})$ is the Fisher information of item $i$ for $\hat{\theta}$ and $B_q$ defines the subset of items that have not yet been presented at the $q$th step of the CAT process [15]. Some additional restrictions may be imposed to $B_q$, such as item exposure or content controls.

The Fisher information function of the three-parameter logistic model is computed as

$$I(\theta) = \frac{2.89a^2(1-c)}{(c + e^{1.7a(\theta-b)})(1 + e^{-1.7a(\theta-b)})^2}. \tag{2}$$

Item information functions are additive and typically combined into the test information function. As the CAT progresses toward, the test information is updated. Let $I_q$ denote the information accumulated at the $q$th step of the CAT process. This information is computed as:

$$I_q(\theta) = \sum_{i=1}^{n} x_i I_i(\theta), \tag{3}$$

where $n$ is the item bank size, and $x_i$ indicates whether or not the item has been administered. Importantly, Fisher information and measurement error are inversely related. Specifically, the measurement error of $\theta$ is asymptotically equal to $I_q(\theta)^{1/2}$ [16].

Previous research has pointed out some limitations of this ISR (e.g., [17,18]). Regarding the accuracy of the trait level estimates, this ISR relies on a punctual estimation of the trait level (i.e., $\hat{\theta}$) to select the next item, it suffers from the problem of possible multiple maxima, and it focuses on differentiating between close trait levels [17]. It should be noted that even though the implications of these limitations decrease as the number of items administered increases, one of the reasons for using a CAT administration is to obtain accurate results in the shortest possible time. This being so, different alternatives to PFI have been proposed, including the KL information [19]. This ISR is a global measure that will be one of the focuses of this paper, and is described in the next section.

**Global measures as alternatives.** PFI consists of selecting the next item so that the Fisher information at $\hat{\theta}_q$ is maximized, where $q$ is the current step of the CAT. Thus, the appropriateness of this ISR depends on how close is $\hat{\theta}_q$ to the true latent trait level, denoted by $\theta$. However, at the early stages of the CAT the deviation of $\hat{\theta}_q$ from $\theta$ may well be large due to the lack of information available. Fisher information is the discrimination power between two close $\theta$ values [19]. This implies two limitations for PFI. First, that the discrimination power between distant $\theta$ values is not considered. Second, that the selection rule implicitly assumes that the deviation of $\hat{\theta}_q$ from $\theta$ is small.

KL, as a global measure of information, on the contrary, considers the information content in the item with respect to a broad range of latent trait levels, addressing the first limitation. Considering that probably $\theta$ is not uniformly distributed, the KL selection rule is typically weighted by the likelihood function or the posterior distribution. Thus, the next item to be selected is determined as

$$j = \arg\max_{i \in B_q} \int_{-\infty}^{\infty} KL_i(\theta \| \hat{\theta}) W(\theta) d\theta, \tag{4}$$

where $W(\theta)$ is equal to the likelihood function, $L(\theta)$, when maximum-likelihood estimation is used. In the case of Bayesian estimation, the weighting function is defined as:

$$W(\theta) = \frac{f_0(\theta) L(\theta)}{\int_{-\infty}^{\infty} f_0(\theta) L(\theta) d\theta}, \tag{5}$$

where $f_0(\theta)$ denotes the prior distribution of $\theta$.

In Eq 4, $KL_i(\theta\|\hat{\theta})$ is calculated as follows:

$$KL_i(\theta\|\hat{\theta}) = P_i(\hat{\theta})\ln\left[\frac{P_i(\hat{\theta})}{P_i(\theta)}\right] + [1 - P_i(\hat{\theta})]\ln\left[\frac{1 - P_i(\hat{\theta})}{1 - P_i(\theta)}\right], \tag{6}$$

where $P_i(\theta)$ is the probability of success on item $i$ for $\theta$. The solution to these integrals is approximated using quadrature points. It should be noted that KL performance is still dependent on the proximity between the estimated trait level and the true trait level.

## New item selection rules for IRT-CAT

As mentioned earlier, the impossibility of using Fisher information in the CD-CAT framework has led to the development of alternative ISRs. These new ISRs include two modifications of the KL method, namely hybrid KL and posterior weighted KL (PWKL) [8]. Recently, Kaplan et al. [10] introduced other two new ISRs. One of them was an improved version of PWKL (MPWKL), and the other one was the generalized deterministic inputs, noisy "and" gate (G-DINA) model discrimination index (also referred to as global-discrimination index [GDI]). MPWKL and GDI were both preferable to PWKL, and yielded highly similar accuracy rates to one another. In the following we describe these two ISRs, adapting the formulation to that of the traditional IRT framework.

**Double KL.** A similar ISR to PWKL was introduced by Chang and Ying [19] in the IRT context, computed as a KL index weighted by the posterior distribution. PWKL (KL hereafter) uses only one integral, whereas Kaplan et al.'s [10] MPWKL (DKL hereafter) uses two integrals. This idea of using two integrals was already briefly mentioned in the discussion of Chang and Ying' study (see Equation 29), but they considered KL within an interval around $\hat{\theta}_q$ instead of KL weighted by posterior. This ISR, which we will call double KL within intervals (DKLI) was defined as:

$$j = \arg\max_{i \in B_q} \int_{\hat{\theta}_q - \eta_q}^{\hat{\theta}_q + \eta_q} \int_{\hat{\theta}_q - \delta_q}^{\hat{\theta}_q + \delta_q} KL_i(\theta_1\|\theta_2)d\theta_1 d\theta_2, \tag{7}$$

where $\eta_q$ and $\delta_q$ determine the amplitude of the interval around $\hat{\theta}_q$. This amplitude decreases with each new administered item, as the uncertainty about the estimated trait level is supposed to be reduced as the test advances.

In the ISR that is proposed in this article–DKL–the next item to be selected is given by

$$j = \arg\max_{i \in B_q} \int \int_{-\infty}^{\infty} KL_i(\theta_1\|\theta_2)W(\theta_1)W(\theta_2)d\theta_1 d\theta_2. \tag{8}$$

DKLI differs in three relevant points with respect to DKL. First, DKLI still needs the computation of an estimated trait level after each new administered item (i.e., $\hat{\theta}_q$). By contrast, in DKL does not uses the estimated trait level for item selection. Second, in DKLI all the computed KL values are equally weighted. With Bayesian KL or KL weighted by likelihood (Eq 4) or with DKL, KL values are weighted by the best available evidence, that is, likelihood function or posterior distribution. Third, DKLI does not consider the potentially relevant information that is outside the interval. With Bayesian KL and KL weighted by likelihood, all the possible pairs of values with respect to the estimated trait level are considered. For DKL, all the possible pairs of values are considered. Previous studies have shown that KL weighted by likelihood reduced measurement error when compared with KL based on (a single) interval [17]. We considered that these were solid reasons for not including DKLI in this study. The main advantage of this DKL is that it does not use the estimated latent trait at all. This is a

convenient characteristic considering that this estimate can be very noisy in the early stages of the CAT. The best way of dealing with this undesirable fact is to use an ISR that does not use those noisy estimates.

**G-DINA model discrimination index.** Kaplan et al. [10] also proposed GDI as an alternative ISR in the context of CD-CAT. The GDI is a discrimination index that was proposed by de la Torre and Chiu [20] as the basis for empirical Q-matrix validation. This index measures the weighted variance of the probability of success of an item given a particular latent trait distribution. The next item to be selected by the adaptive algorithm is the one with the highest GDI:

$$j = \arg\max_{i \in B_q} GDI = E\{VAR[P_i(\theta)]\} = \int_{-\infty}^{\infty} P_i^2(\theta) W(\theta) - [\int_{-\infty}^{\infty} P_i(\theta) W(\theta)]^2 d\theta. \qquad (9)$$

It can be noted from this equation that the estimated trait level is either used with this method.

**Examples with some fictitious items.** Figs 1 and 2 illustrate DKL and GDI, respectively, for two fictitious items at different stages of the CAT. KL measures the discrepancy between two probability distributions. The size of KL will be larger the more different the response distributions associated to the two levels $\theta$ that are being compared. This can be observed in Fig 1. Item B has a much higher $a$ parameter and this is indicated by a higher peak at [$\theta_1 = -4$, $\theta_2 = 4$] compared to the representation of item A. For the same levels of $a$ parameter, the effect of $c$ would be also reflected on this height. The height of the peak is important because DKL is computed as the sum of all the depicted values weighted by the likelihood function or posterior distribution. As the CAT progresses, some of the trait levels will be more plausible and the area determining the size of DKL will be smaller. In the example depicted, at item position 3 items A and B have a DKL associated of 0.68 and 0.71, respectively. The item with the highest $a$ parameter (i.e., item B) is preferred. At item position 21, items A and B have a DKL associated of 0.18 and 0.25, respectively. Again, item B is preferred. Compared to DKL, KL will only compare one specific trait level (i.e., $\hat{\theta}$) with all the other possible values (i.e., $\theta = [-4,4]$ in this example). This would be computed in Fig 1 considering all the KL values that are defined by $\theta_1 = \hat{\theta}$.

GDI is illustrated in Fig 2. This index is computed as the weighted variance of the probability of success. We can note from Fig 2 that this index might be more affected by the $c$ parameter given that it defines the lower asymptote of the item response function thus determining the range of values. The range of values is indeed a common measure of variance. In this example, the items selected according to GDI at item position 3 and 21 differ. At item position 3, items A and B have associated GDI values of 0.112 and 0.084, respectively. Thus, the item with the lowest $c$ parameter is preferred even though it has a much smaller $a$ parameter. On the contrary, at item position 21 item B would be preferred because it has a greater GDI value associated (0.044) compared to item A (0.037). The reason for this is that the weighting function that was used in the example was centered at $\theta = 0$. Thus, the large difference in $a$ parameter made this item more preferable at a late stage of the CAT.

## Trait level estimation

The estimation of the latent trait level is generally based on observed response pattern. The basic idea underlying the estimation procedures is finding the trait level that maximizes the probability of the observed pattern of responses. This is the most basic and often used estimation procedure, referred to as maximum likelihood (ML) estimation. The likelihood function

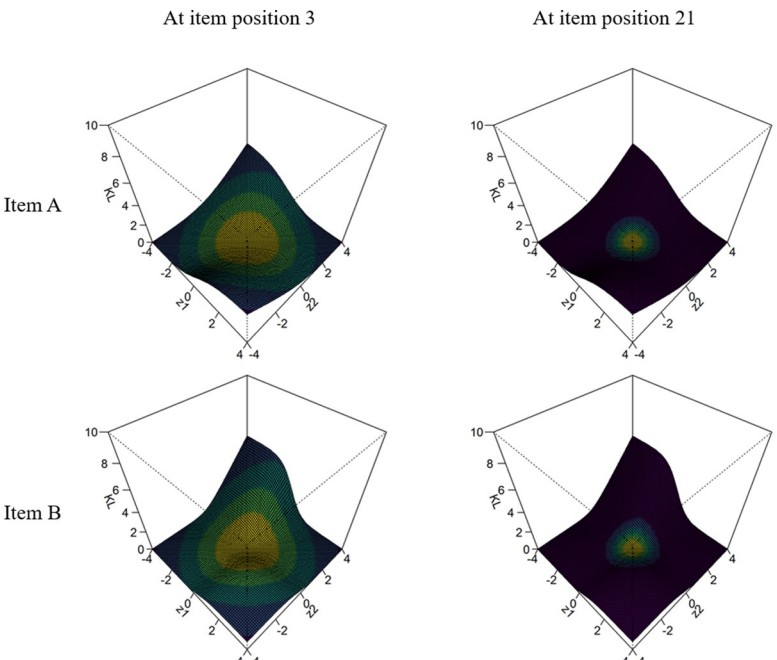

**Fig 1. DKL for two fictitious items.** Both items share the same $b$ parameter ($b_A = b_B = 0$), but differ in their discrimination ($a_A = 1$; $a_B = 2$) and pseudo-guessing ($c_A = 0$; $c_B = 0.30$) parameters. Two item positions are illustrated: Item position 3 and 21. At item position 3, the examinee has one correct and one incorrect response. At item position 21, the examinee has ten correct and ten incorrect responses. The color gradient represents the weight to be applied to the KL and DKL computation based on the examinee's likelihood function. $\theta$ is represented as $z$.

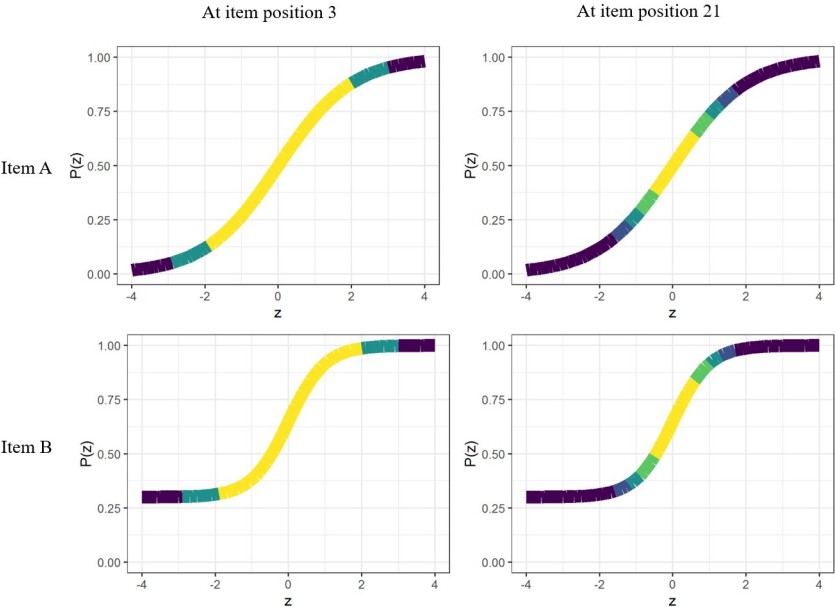

**Fig 2. GDI for two fictitious items.** Both items share the same $b$ parameter ($b_A = b_B = 0$), but differ in their discrimination ($a_A = 1$; $a_B = 2$) and pseudo-guessing ($c_A = 0$; $c_B = 0.30$) parameters. Two item positions are illustrated: Item position 3 and 21. At item position 3, the examinee has one correct and one incorrect response. At item position 21, the examinee has ten correct and ten incorrect responses. The color gradient represents the weight to be applied to the GDI computation based on the examinee's likelihood function. $\theta$ is represented as $z$.

is obtained as indicated in:

$$L(\theta, \mathbf{x}, \mathbf{g}) = \prod_{i=1}^{n} [P_i^{g_i}(\theta)(1 - P_i(\theta))^{1-g_i}]^{x_i}, \qquad (10)$$

where for every examinee's response $g_i$ = 0 or 1 (0 denotes an incorrect response and 1 denotes a correct response). Alternatively, some Bayesian methods [21,22] have been proposed. These methods combine the evidence contained in the response patterns with prior knowledge of the probability distribution of the latent trait in the population of respondents. Wang and Vispoel [23] compared some of these estimation methods in a simulation study using CATs, and found that the *expected a posteriori* (EAP) method was the best performing method among the Bayesian methods. This previous study explored differences in terms of bias, standard error (SE), and root mean squared error (RMSE). According to their results, ML provided lower bias for a moderate discrimination item bank, but a higher standard error and RMSE. Overall, ML and EAP differences in terms of bias were generally smaller when the item bank discrimination was high. With less than 30 items administered, EAP led to a lower bias in this condition. In selecting the items, Wang and Vispoel used PFI as ISR. The prior study by van der Linden [24] also suggested that Bayesian item selection criteria might be superior to PFI with ML estimation of the trait level. However, it remains unknown if these differences generalize across different ISRs. Besides, it would be worth exploring differences in other dependent variables such as test overlap and features of the items administered in order to gain a better understanding of these trait level estimators. The overlap rate has been used as an indicator of item bank security. It is an estimate of the proportion of administered items that are shared by two examinees [25].

## Goal of the present study

The goal of the present study is twofold. The first goal of the study is to evaluate how the two ISRs originally proposed within the CD-CAT framework (i.e., DKL and GDI) perform in the traditional CAT framework. For comparison purposes, the two most commonly used rules in the IRT context (i.e., PFI and KL) are also considered. These ISRs are compared in terms of accuracy and security, and the pattern of items selected with each ISR is explored. The application of CDMs to adaptive testing is a relatively new area of research which has been aided by the experiences in the traditional IRT context. This study is pioneering in the sense that this is probably one of the first times that CDM methodologies are being exported to the traditional IRT context. The second goal is to explore the effect of the trait level estimation method on the performance of the CATs. This research extent previous research (e.g., [23,24]) by incorporating different ISRs and dependent variables. We expected (a) DKL and GDI to outperform the other ISRs because these new proposals do not use the estimated trait level for item selection, and (b) that these differences will be smaller for longer test cases [17,26].

## Method

We used a code written in Pascal to conduct CAT simulations under different conditions in order to compare the ISRs. In Eqs 4 and 8 and 9 and for the trait level estimators, integrals were approximated using 81 quadrature points. The number of quadrature points was set to be large enough to provide accurate results, not to be computationally efficient. The following are details of the simulation study. The weighting function $W(\theta)$ was equal to the likelihood function when ML estimation was used, and equal to the posterior distribution when EAP was used. The prior distribution of $\theta$ (i.e., $f_0(\theta)$) was set to U[−4, 4]. As the prior was uniform, there

were, in fact, no differences between $W(\theta)$ by latent trait estimator. The open database and code files are available at the Open Science Framework repository (https://osf.io/kagxy/).

## Item banks and test length

Ten item banks were constructed, each of them containing 500 items. Item parameters were generated randomly from the following distributions: for $a \sim N(1.20, 0.25)$, $b \sim N(0, 1)$, and $c \sim N(.25, .02)$. The maximum number of items to be administered to each examinee was set to 20.

## Trait level of the simulees

We were interested in obtaining the results for the overall population and conditional on several different $\theta$ values. For overall results, 5,000 values were generated from a standard normal distribution. This process was repeated for each of ten item banks. For conditional results, we used nine trait level values, ranging from –2 to 2 in steps of 0.5, with 1,000 examinees per trait level and item bank. The total number of examinees used in this study was $10 \times (5,000 + 9,000) = 140,000$.

## Starting rule

The starting $\hat{\theta}$ was sampled at random from the interval (–0.5, 0.5). The likelihood function that is used in all the ISRs but PFI requires a response pattern. In order to apply these ISRs with effect from the very first item two fictitious items were used. This strategy, used for example in Barrada et al.[17], consists in considering one correct and one incorrect response to two items with the same characteristics ($a = 0.5$, $b = \hat{\theta}_0$, and $c = 0$). These two responses are only used in this step of the CAT process, but it is important to consider this when interpreting the results for the first items.

## Trait estimation

We compared ML estimation and EAP estimation. Dodd's [27] approach for dealing with constant patterns in ML was applied until the examinee obtained correct and incorrect responses. This approach consists of increasing $\hat{\theta}$ by $(b_{max} - \hat{\theta})/2$ when all the responses are correct; and decreasing $\hat{\theta}$ by $(\hat{\theta} - b_{min})/2$ when all the responses are incorrect. The parameters $b_{max}$ and $b_{min}$ correspond to the maximum and minimum $b$ parameter in the item bank, respectively. This needs to be also considered when interpreting the results for the first items. With EAP estimation a uniform prior over [–4, 4] was used (e.g., [28–30]). For both estimators $\hat{\theta}$ was estimated within the interval [–4, 4].

## Performance measures

Six dependent variables were used to evaluate the performance of the ISRs. Results were computed for each one of the ten different item banks and averaged across them to obtain more stable results. To evaluate measurement accuracy, bias and RMSE were considered:

$$RMSE = \left( \sum_{h=1}^{m} (\hat{\theta}_h - \theta_h)^2 / m \right)^{1/2}, \tag{11}$$

$$Bias = \sum_{h=1}^{m} (\hat{\theta}_h - \theta_h)/m, \tag{12}$$

where $m$ is the number of examinees, $\hat{\theta}_h$ is the estimated trait level for the $h$-th examinee and $\theta_h$ is the real trait level. The overlap rate was computed as indicator of test security [31]:

$$T = \frac{n}{q}S^2_{er} + \frac{q}{n},\qquad(13)$$

where $S^2_{er}$ is the variance of the exposure rates of the items. We also computed the mean values of the $a$ and $c$ parameters of the administered items in order to describe the characteristics of items selected by each ISR. The correlations between the item exposure rates were also computed. This is an indicative of the convergence among the ISRs. Finally, the time required to select a single item for each ISR was recorded.

## Results

In the following we describe the results for the average estimates of the dependent variables across the ten item banks. The variability across the item banks was generally small and always negligible when the CAT length was equal or greater than four and/or the trait level estimator was EAP. Thus, although presented here, it is important to note that the results at the very start of the CAT (1–3 items) should be interpreted with caution. The reason is that the ISRs might be affected differently by some specific characteristics of the CAT algorithm at the beginning of the test (e.g., Dodd's method, the use of two fictitious items in the likelihood computation for the first item).

Fig 3 summarizes the RMSE and overlap rate results conditional on the number of items presented. We can see from the figure that, logically, as the number of items presented was increased, RMSE decreased. This became less noticeable as the CAT progressed. There were only slight differences between ML and EAP regarding RMSE. For example, for a 10-item CAT, RMSE mean values across the four ISRs were 0.469 and 0.449 for ML and EAP, respectively. Specifically, when the ML estimator was employed, it was observed that after the administration of five items, KL, DKL, and GDI were consistently better than PFI. The performance of all ISRs was more similar when the EAP estimation method was used. In this condition, DKL was generally the best performing method and PFI was not always the worse performing

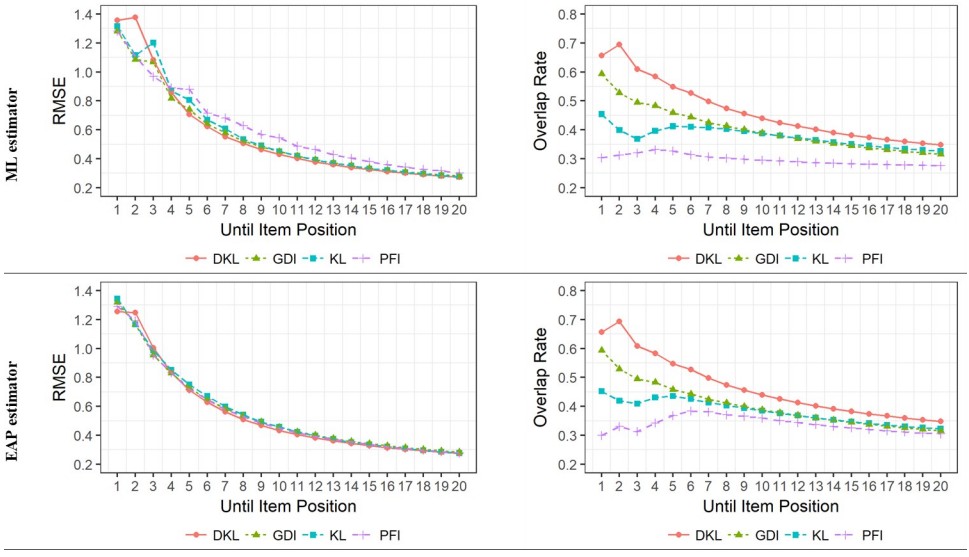

**Fig 3. RMSE and overlap rate according to item selection rule, item position, and trait level estimator.**

method. Overall, for both estimation methods, DKL was better than GDI, and they both were generally better than KL and PFI.

RMSE for DKL was smaller than 0.50 for CAT lengths of nine items for both ML and EAP estimators. The results for this specific test length are detailed in the following to illustrate the overall description provided above. When the ML/EAP estimator was employed, RMSE values for a 9-item CAT were 0.568/0.484 (PFI), 0.492/0.493 (KL), 0.463/0.469 (DKL), and 0.484/0.492 (GDI). For tests longer than 17 (EAP estimator) and 20 (ML estimator) items, there appeared to be no differences among the ISRs.

Regarding the overlap rate results, it was observed that PFI showed the best test security, with an overlap rate always smaller than .40, and did not dramatically change as the CAT progressed. When the EAP estimator was used, the overlap rate for PFI was slightly higher. All the rules alternative to PFI showed an overlap rate higher than that obtained with PFI. At the beginning of the CAT, DKL showed the highest overlap rate, followed by GDI, but the differences between the ISR became smaller as the number of items administered increased. As we have already described, the differences in RMSE were minimal, although they could be still detected for DKL with a test length of 20 items. In comparison, the differences in overlap rate were larger. Better accuracy was achieved with a cost in overlap. It is important to remark that for CATs longer than 13 items, all the ISRs had overlap rates lower than .40.

As a mean to better understand the overall accuracy results, the conditional results for bias and RMSE are shown in Table 1. Results for a for a 10-item CAT are described in the following. Overall, DKL performed the best and PFI performed the worse in terms of bias. KL generally had a good performance when the ML estimator was employed. However, it got even worse than PFI when the EAP estimator was used. The pattern of results for RMSE was similar to that of the bias. DKL performed the best overall. This was particularly noticeable for low trait levels. The bias and RMSE decreased as the number of items increased. As can be seen in the table, bias values were generally close to 0, and RMSE was always smaller than 0.45 when the CAT ended (i.e., 20 items). The differences among the methods became less noticeable, but generally indicated gains in accuracy for the DKL and GDI as indicated by the overall average of the absolute values.

Fig 4 shows the average $a$ and $c$ parameters of the presented items. At the beginning of the test, all ISRs tended to select items with the $a$ parameter clearly above the mean of this parameter in the item bank (i.e., 1.2). Subsequently, and probably because highly discriminating items were exhausted, items with lower $a$ parameters were progressively administered. Comparing the different ISRs indicates that at the beginning of the CAT, DKL used items with the highest $a$ parameter, and GDI with the lowest $a$ parameter. Differences among ISRs became negligible when the number of administered items was higher than five, with the exception of PFI when the ML estimator was used that was only similar to the others when more than ten items were administered. Regarding the $c$ parameter, the ISRs tended to administer items with a value in this parameter below the mean of this parameter in the item bank (i.e., .25). The average in this parameter increased in later stages of the CAT process, probably because items with low $c$ parameter were already exhausted. In comparison to PFI, the other ISRs selected items with a smaller $c$ parameter at the beginning of the CAT. This tendency was much more pronounced for GDI. There were no differences in the performance of PFI, DKL, and GDI combined with the ML and EAP estimators. On the contrary, KL combined with the ML estimator used items with lower $c$ parameter at the beginning of the CAT, compared to KL combined with the EAP estimator. Differences among ISRs became negligible when the number of administered items was higher than approximately ten.

Finally, Fig 5 depicts the correlations between the item exposure rates. The different ISRs selected different items at the beginning of the CAT, and became more similar as the CAT

**Table 1. Conditional Bias and RMSE According to Item Selection Rule and Trait Level Estimator for a 10-item and 20-item CAT.**

| | | | $\theta$ | | | | | | | | | |
|---|---|---|---|---|---|---|---|---|---|---|---|---|
| **10 items** | | | **Bias** | | | | | | | | | |
| | | ISR | -2 | -1.5 | -1 | -0.5 | 0 | 0.5 | 1 | 1.5 | 2 | Row Mean* |
| | ML Estimator | PFI | 0.18 | 0.14 | 0.08 | 0.08 | 0.08 | 0.08 | 0.06 | 0.04 | 0.05 | 0.0864 |
| | | KL | 0.04 | 0.04 | 0.03 | **0.02** | **0.02** | **0.03** | **0.02** | 0.03 | **0.03** | 0.0300 |
| | | DKL | **0.03** | **0.03** | **0.03** | 0.03 | 0.03 | 0.03 | 0.03 | **0.02** | 0.03 | **0.0297** |
| | | GDI | 0.06 | 0.06 | 0.04 | 0.06 | 0.05 | 0.05 | 0.03 | 0.04 | 0.04 | 0.0489 |
| | EAP Estimator | PFI | -0.06 | -0.06 | -0.05 | -0.04 | -0.04 | -0.02 | -0.03 | -0.02 | 0.01 | 0.0366 |
| | | KL | -0.06 | -0.06 | -0.07 | -0.07 | -0.06 | -0.04 | -0.04 | -0.03 | 0.01 | 0.0506 |
| | | DKL | **-0.06** | **-0.04** | **-0.02** | **-0.01** | **-0.01** | **0.01** | **0.00** | **0.00** | 0.03 | **0.0224** |
| | | GDI | -0.06 | -0.05 | -0.04 | -0.04 | -0.04 | -0.03 | -0.02 | -0.03 | **0.01** | 0.0367 |
| | | | **RMSE** | | | | | | | | | |
| | | ISR | -2 | -1.5 | -1 | -0.5 | 0 | 0.5 | 1 | 1.5 | 2 | Row Mean* |
| | ML Estimator | PFI | 0.85 | 0.74 | 0.65 | 0.57 | 0.49 | 0.43 | 0.40 | **0.39** | **0.41** | 0.5484 |
| | | KL | 0.56 | 0.50 | 0.48 | 0.44 | 0.44 | 0.42 | 0.41 | 0.41 | 0.41 | 0.4529 |
| | | DKL | **0.49** | **0.45** | **0.44** | **0.42** | **0.42** | **0.42** | 0.42 | 0.43 | 0.44 | **0.4357** |
| | | GDI | 0.57 | 0.53 | 0.47 | 0.45 | 0.43 | 0.42 | **0.40** | 0.41 | 0.43 | 0.4572 |
| | EAP Estimator | PFI | 0.54 | 0.49 | 0.45 | 0.46 | 0.44 | 0.43 | 0.43 | 0.44 | 0.47 | 0.4612 |
| | | KL | 0.56 | 0.51 | 0.47 | 0.45 | 0.44 | 0.45 | 0.43 | 0.43 | 0.46 | 0.4677 |
| | | DKL | **0.49** | **0.46** | **0.45** | **0.42** | **0.42** | **0.42** | **0.42** | 0.44 | 0.47 | **0.4449** |
| | | GDI | 0.56 | 0.50 | 0.48 | 0.44 | 0.45 | 0.44 | 0.43 | **0.43** | 0.46 | 0.4653 |
| **20 items** | | | **Bias** | | | | | | | | | |
| | | ISR | -2 | -1.5 | -1 | -0.5 | 0 | 0.5 | 1 | 1.5 | 2 | Row Mean* |
| | ML Estimator | PFI | 0.03 | 0.04 | 0.02 | 0.02 | 0.03 | 0.03 | 0.03 | 0.02 | 0.03 | 0.0271 |
| | | KL | 0.00 | 0.01 | 0.01 | 0.01 | 0.01 | 0.01 | **0.01** | 0.02 | **0.02** | 0.0136 |
| | | DKL | **0.00** | **0.01** | **0.01** | **0.01** | **0.01** | **0.02** | 0.01 | **0.02** | 0.03 | **0.0126** |
| | | GDI | 0.01 | 0.02 | 0.02 | 0.03 | 0.03 | 0.03 | 0.02 | 0.03 | 0.03 | 0.0235 |
| | EAP Estimator | PFI | **-0.07** | -0.04 | -0.02 | -0.01 | -0.01 | 0.01 | **0.00** | 0.01 | 0.03 | 0.0202 |
| | | KL | -0.08 | -0.04 | -0.03 | -0.01 | -0.01 | **0.00** | 0.00 | 0.01 | 0.03 | 0.0232 |
| | | DKL | -0.08 | -0.03 | **-0.02** | **-0.01** | **0.00** | 0.00 | 0.01 | 0.01 | 0.03 | 0.0206 |
| | | GDI | -0.07 | **-0.03** | -0.02 | -0.01 | -0.01 | 0.00 | 0.00 | **0.00** | 0.02 | **0.0190** |
| | | | **RMSE** | | | | | | | | | |
| | | ISR | -2 | -1.5 | -1 | -0.5 | 0 | 0.5 | 1 | 1.5 | 2 | Row Mean* |
| | ML Estimator | PFI | 0.45 | 0.38 | 0.33 | 0.29 | 0.27 | 0.26 | **0.26** | **0.26** | 0.29 | 0.3091 |
| | | KL | 0.34 | 0.30 | 0.28 | 0.26 | 0.26 | 0.26 | 0.26 | 0.27 | **0.28** | 0.2795 |
| | | DKL | **0.33** | **0.29** | **0.27** | **0.26** | **0.26** | **0.26** | 0.27 | 0.28 | 0.29 | **0.2776** |
| | | GDI | 0.35 | 0.31 | 0.28 | 0.27 | 0.27 | 0.27 | 0.27 | 0.27 | 0.29 | 0.2857 |
| | EAP Estimator | PFI | 0.37 | 0.31 | 0.28 | 0.27 | 0.26 | 0.26 | 0.26 | 0.28 | 0.31 | 0.2892 |
| | | KL | 0.37 | 0.31 | 0.28 | 0.27 | 0.27 | **0.26** | 0.26 | **0.27** | 0.30 | 0.2892 |
| | | DKL | **0.36** | **0.30** | **0.28** | **0.26** | **0.26** | 0.26 | **0.26** | 0.28 | 0.31 | **0.2856** |
| | | GDI | 0.38 | 0.32 | 0.29 | 0.27 | 0.27 | 0.27 | 0.27 | 0.28 | **0.30** | 0.2948 |

*Note*. ISR: Item selection rule.

*: Row mean of absolute values. A grey gradient is used to facilitate the visualization of the data. Within each section of the table, grey cells indicate low bias or RMSE, whereas white cells indicate high bias or RMSE. Minimum values within each column are shown in bold.

progressed. The ISR most differentiated from the others in its item selection patterns is GDI, especially at the early stages of the CAT. This result is congruent with what was shown in Fig 4.

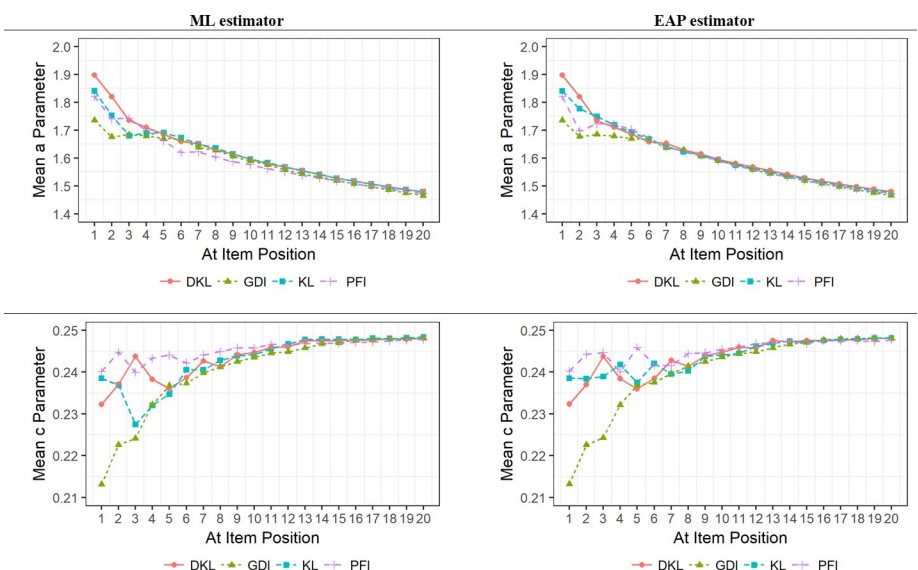

**Fig 4. Mean *a* and *c* parameters of the administered items according to item selection rule, item position, and trait level estimator.**

PFI and GDI were the ones more different from each other. GDI and DKL followed a clearly different path at the beginning of the CAT. KL and DKL were generally the ones more similar to each other. Nonetheless, it must be noted that the coincidence in the item selection patterns was generally high, even with ten items administered with mean correlations higher than .90 for both ML and EAP estimators. As can be seen from Fig 5, even for the two ISR with the most different pattern (i.e., GDI and PFI) the correlation between their item exposure rates was approximately .84 with ten items presented. There were only minor some differences regarding the trait level estimator. GDI was less similar to KL when the trait level estimator was EAP.

The code was run on a computer with processor of 3.0 GHz. The average times, in milliseconds, for selecting each item were 0.3 ms/item for PFI, 11.2 ms/item for KL, 3.9 ms/item for GDI, and 219.4 ms/item for DKL. As the computation time is dependent among many computer characteristics, we also computed the ratio, relative to PFI: KL was 36.5 times slower, 12.6 slower for GDI, and 713.3 times slower for DKL.

## Discussion

The present study aimed to examined the measurement accuracy and test security of two new ISRs, namely, DKL and GDI. These two algorithms were originally developed in the CDM context, where they were proven to be useful item selection methods [10]. We thus assessed the bias, RMSE, overlap rate, mean values of the *a* and *c* parameters administered, and the correlation between the item exposure rates. The previously best available implementation of KL and the gold standard in adaptive testing, that is, PFI, were included for comparison purposes.

Both DKL and GDI were shown to be more accurate than the traditional ISRs, with DKL providing slightly better results. Interestingly, DKL and GDI obtained high levels of accuracy through different ways–at the beginning of the test, the GDI tended to use items with the smallest possible *c* parameters, whereas DKL, and KL and PFI, to a smaller degree, tended to use items with the highest possible *a* parameters. The dissimilarity in the item usage was evident in the small correlation between the item exposure rates of the two new methods. These

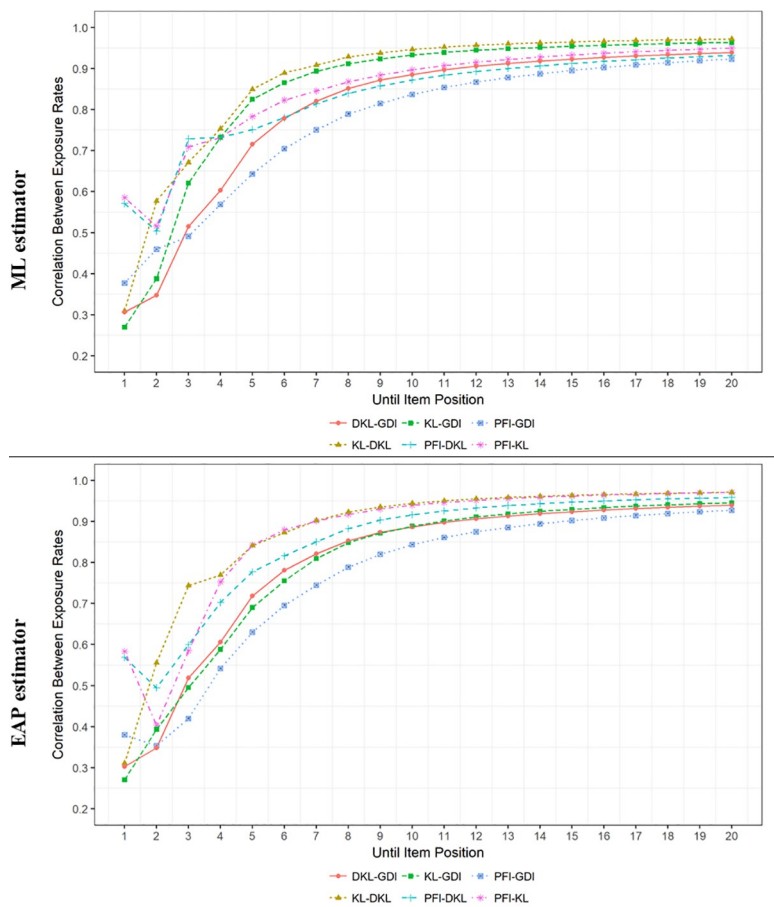

**Fig 5. Correlation between the item exposure rates of the different item selection rules according to item position and trait level estimator.**

results indicate that contingent on the ISR, item selection will depend on the *c*, not only on the *a* parameters, as it has been generally acknowledged [32]. In contrast to the results in the CDM context where differences between DKL and GDI were negligible [10], DKL generally provided higher accuracy, which was achieved at the expense of lower test security. Compared to the new ISRs, KL and PFI had lower overlap rates. Nonetheless, all the ISRs considered in this work had acceptable overlap rates for tests longer than ten items or so [25].

Based in the simulated conditions, ten items might be a good fixed-length test for balancing accuracy and test security. As expected, improvement in accuracy became less prominent as the number of administered items increased. Admittedly, the proposed ISRs, particularly DKL, were slower than PFI; however, the computation time per item was always markedly below a second so this differences may be deemed as trivial in any applied context. It should be noted that the simulation code was not written to optimize execution times. For example, DKL, with its double integration, used $81 \times 81 = 6561$ quadrature points. We expect that this number could be reduced with a negligible loss in accuracy.

All things considered, the new ISRs can provide the same levels of accuracy with fewer items administered, which is of major importance in contexts like educational or medical testing, where testing time is always an issue. Along this line, patient-reported outcomes (PROs) are, and will be even more important in the future clinical world [33]. Reducing testing time in

medical evaluation is of crucial importance. Accordingly, systems like PROMIS or CAT-5D-QOL were developed in response to the request for a more efficient testing environment [34,35]. Electronic PROs have been found to have some advantages over paper based PROs, including avoiding data entry errors and an immediate access to data, and have been successfully implemented in some applications (e.g., [36]). Regarding the educational context, the efficiency of adaptive testing can make regular classroom assessment less intrusive, thus more practicable, and will allow to teachers to devote more time to instruction [37]. DKL and GDI improvement's in terms of accuracy was found to be higher at the beginning of the CAT, when only a few items have been administered. This is of practical importance because instruments in these contexts should ideally contain as few optimal items as possible. Importantly, these new methods are easy to implement, and GDI is already available in the catR package of R [38].

Another important contribution of the present study is that ML and EAP trait level estimation methods were systematically compared. We found that EAP was generally better in the first stages of the CAT, and provided the same results as ML when a medium to large number of items were administered. As in the study of Wang and Vispoel [23], we found that ML did not provide consistently a lower bias than EAP, but it generally provided a higher RMSE. On the other hand, we found that the trait level estimation method and the ISRs interacted. The different ISRs were more similar when the EAP estimator was employed. Consistent with previous research, we found that PFI and KL did not differ in terms of RMSE when the EAP estimator was used [26]. In contrast, we observed that KL consistently provided a lower RMSE compared to that of PFI when the ML estimator was used. This result is also in line with previous research [17,18].

To keep the scope of this study manageable, a few simplifications about factors affecting the CAT performance were made. These included focusing on the fixed CAT-length condition and the unconstrained CAT, where exposure control was not considered. There might be a trade-off between accuracy and security for the different ISRs that future studies should consider. As was found in previous studies comparing different ISRs (e.g., [17,18]), PFI was the rule with the greatest measurement error and the smallest overlap. At the other extreme was DKL, which showed a high overlap rate, but with a higher accuracy. This is possible due to the fact that PFI relies on the current trait level estimate, and try to find items with $b$ parameters very close to that current estimate; in contrast, global measures do not rely on a single estimate. As exposure control becomes more necessary as the stakes get higher, future research should explore how item exposure control method can be implemented with DKL and GDI to improve test security without sacrificing much accuracy. Different studies have shown that it is possible to improve overlap rate with minimal impact on accuracy (e.g., [39,40]). A more thorough comparison of rules that differ in terms of RMSE and overlap rate at the same time require an extensive manipulation of the maximum exposure rates (e.g., [18]). Another possible way to reduce test overlap would be increasing item bank size. Automatic item generation has the potential to help in this respect [41]. In this study we used true item parameter values. In other words, we assumed that operational item parameters were estimated without any error. In practice, this would not be the case. Further research should explore the impact of calibration error with different ISRs [42–44]. This study focused on the ML and EAP estimators because of their popularity and availability (e.g., [38]). This study can be extended in the future by comparing different trait level estimators such as the essentially unbiased estimators proposed by Wang, Hanson, and Lau [45]. Finally, a natural direction would be to extend the benefits of the proposed ISRs to multidimensional CAT. There exists a prior work by Wang, Chang, and Boughton [46] where the authors extend the KL index to the multidimensional

case. This can be, however, computationally expensive for DKL and GDI because such an extension would require integrating across multiple dimensions.

## Author Contributions

**Conceptualization:** Juan R. Barrada, Jimmy de la Torre.

**Formal analysis:** Miguel A. Sorrel, Juan R. Barrada, Francisco José Abad.

**Funding acquisition:** Francisco José Abad.

**Methodology:** Juan R. Barrada, Jimmy de la Torre.

**Project administration:** Miguel A. Sorrel.

**Software:** Juan R. Barrada.

**Supervision:** Francisco José Abad.

**Visualization:** Miguel A. Sorrel.

**Writing – original draft:** Miguel A. Sorrel.

**Writing – review & editing:** Miguel A. Sorrel, Juan R. Barrada, Francisco José Abad.

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
