## [Decision Letter · Decision Letter 0]

8 Aug 2019

PONE-D-19-15008

Exchanging Item Selection Rules from Cognitive Diagnosis Modeling to Traditional Item Response Theory in Computerized Adaptive Testing

PLOS ONE

Dear Dr. Barrada,

Thank you for submitting your manuscript to PLOS ONE. After careful consideration, we feel that it has merit but does not fully meet PLOS ONE’s publication criteria as it currently stands. Therefore, we invite you to submit a revised version of the manuscript that addresses the points raised during the review process.

I have received three reviews of your manuscript from experts in the field. As you can see, they see merit in your work but suggest several modifications. I will not reiterate the specific points given by the reviewers because they speak for themselves. However, I encourage you to address each of them carefully and submit a revised version of your manuscript.

We would appreciate receiving your revised manuscript by Sep 22 2019 11:59PM. To enhance the reproducibility of your results, we recommend that if applicable you deposit your laboratory protocols in protocols.io, where a protocol can be assigned its own identifier (DOI) such that it can be cited independently in the future. For instructions see: http://journals.plos.org/plosone/s/submission-guidelines#loc-laboratory-protocols

We look forward to receiving your revised manuscript.

Kind regards,

Timo Gnambs

Academic Editor

PLOS ONE

 "This research was partially supported by Grant PSI2017-85022-P (Ministerio de Ciencia, Innovación y Universidades, Spain)."

Reviewers' comments:

Reviewer's Responses to Questions

**Comments to the Author**

1. Is the manuscript technically sound, and do the data support the conclusions?

Reviewer #1: Partly

Reviewer #2: Yes

Reviewer #3: Yes

2. Has the statistical analysis been performed appropriately and rigorously? 

Reviewer #1: Yes

Reviewer #2: Yes

Reviewer #3: Yes

3. Have the authors made all data underlying the findings in their manuscript fully available?

Reviewer #1: Yes

Reviewer #2: Yes

Reviewer #3: Yes

4. Is the manuscript presented in an intelligible fashion and written in standard English?

Reviewer #1: Yes

Reviewer #2: Yes

Reviewer #3: Yes

5. Review Comments to the Author

Reviewer #1: Overall, I have concerns about the merit of the study—from both the practical and theoretical viewpoints. The authors advocate the use of the double Kullback-Leibler divergence (DKL) and the discrimination index (GDI) as item selection methods in computerized adaptive testing (CAT); however, the overall performances of these measures are somewhat worrisome and I’m not sure if they provide any benefits over the existing methods. Figure 3 shows that the four item selection methods performed very comparably when EAP estimator was applied in estimation of the trait parameters (and thus no clear benefit). When ML estimator was used, DKL performed outperformed marginally after the administration of four items and yet showed the worst performance in the beginning stage of CAT. The authors argued for DKL because simple KL-based item selection can be disturbed by unstable trait estimates (p. 6 linear 140-143). The DKL however seemed to show even more disturbing item selection in the early stage of CAT. The item overlap results also suggest that DKL and GDI tended to suffer from skewed item usage. Overall, it was unclear if the suggested procedures present clear advantages despite the poorer performance in item exposure control and heavier computational load.

The manuscript may be revised by lightening the strong advocacy of DKL and GDI (with the proper explanation of their pros and cons) and by elaborating more on the situations where the suggested methods can lend better utility.

Reviewer #2: This paper proposed two new item selection methods for IRT-based CAT from two CD-CAT-based algorithms one of which demonstrate excellent performance in measurement efficiency and item exposure rates. After careful reading of this manuscript, I recommend its publication in Plosone with some minor revision.

The main problem is that the authors did not explain the similarity/difference in the double KL (DKL) and the existing Bayesian KL in IRT-CAT. It makes more sense to use the Bayesian KL as a baseline method rather than the KL method.

Reviewer #3: This work evaluates the performance of two item selection rules that were originally proposed in the Cognitive Diagnosis (CD)-Computerized Adaptive Testing (CAT) framework, the double Kullback-Leibler information (DKL) and the generalized deterministic inputs, noisy “and” gate model discrimination index (GDI), in the setting of traditional CAT under the unidimensional Item Response Theory (IRT) model. The authors gave a very nice review of the existing works and performed a simulation study to compare the DKL and GDI with two existing methods in CAT. Both the measurement accuracy and test security were compared, and the simulation shows promising results of the two proposed approaches (DKL and GDI).

This is an interesting study and it is well written. As pointed out by the authors, it is one of the first times that CDM methodologies are applied to the traditional IRT context. I have the following specific comments, which I hope would be helpful.

1. As the authors mentioned on Page 6, Chang and Ying also discussed the idea of using two integrals for KL considered in a range around \\hat\\theta. Would the authors comment on how this “locally” double-integral approach compares with the proposed globally integrals in DKL?

2. The evaluation of DKL involves with double integration. Would the authors comment on the computation cost for this method?

3. I noticed from the simulation results that for the DKL method, there seem to have an increment of MSE for the first two items in Figure 3. Is this due to simulation error or is there any other reason for such trend?

4. The performance of the DKL seems worse than KL for the first few items. Is there any intuition why this would happen?

Related to this question, would the performance be further improved if selecting the first few items with KL and then using DKL for later items?

5. As the DKL and GDI were originally proposed for CDMs with multiple attributes, I was wondering if these methods can also be extended to multidimentional IRT models? A discussion would be helpful.

6. PLOS authors have the option to publish the peer review history of their article (what does this mean?). If published, this will include your full peer review and any attached files.

Reviewer #1: No

Reviewer #2: No

Reviewer #3: No

---

## [Author Response · Author response to Decision Letter 0]

14 Oct 2019

Please, see the Cover Letter, where we included our responses.

---

## [Decision Letter · Decision Letter 1]

21 Nov 2019

PONE-D-19-15008R1

Exchanging Item Selection Rules from Cognitive Diagnosis Modeling to Traditional Item Response Theory in Computerized Adaptive Testing

PLOS ONE

Dear Dr. Barrada,

Thank you for submitting your manuscript to PLOS ONE. After careful consideration, we feel that it has merit but does not fully meet PLOS ONE’s publication criteria as it currently stands. Therefore, we invite you to submit a revised version of the manuscript that addresses the points raised during the review process.

I appreciate the thorough revision of your manuscript. I believe it presents an interesting addition to the scientific literature. Therefore, I ask you to address the remaining minor comments raised by Reviewer 3 before I can formally accept you work for publication.

We would appreciate receiving your revised manuscript by Jan 05 2020 11:59PM. To enhance the reproducibility of your results, we recommend that if applicable you deposit your laboratory protocols in protocols.io, where a protocol can be assigned its own identifier (DOI) such that it can be cited independently in the future. For instructions see: http://journals.plos.org/plosone/s/submission-guidelines#loc-laboratory-protocols

We look forward to receiving your revised manuscript.

Kind regards,

Timo Gnambs

Academic Editor

PLOS ONE

Reviewers' comments:

Reviewer's Responses to Questions

**Comments to the Author**

1. If the authors have adequately addressed your comments raised in a previous round of review and you feel that this manuscript is now acceptable for publication, you may indicate that here to bypass the “Comments to the Author” section, enter your conflict of interest statement in the “Confidential to Editor” section, and submit your "Accept" recommendation.

Reviewer #1: All comments have been addressed

Reviewer #3: All comments have been addressed

2. Is the manuscript technically sound, and do the data support the conclusions?

Reviewer #1: Yes

Reviewer #3: Yes

3. Has the statistical analysis been performed appropriately and rigorously? 

Reviewer #1: Yes

Reviewer #3: Yes

4. Have the authors made all data underlying the findings in their manuscript fully available?

Reviewer #1: Yes

Reviewer #3: Yes

5. Is the manuscript presented in an intelligible fashion and written in standard English?

Reviewer #1: Yes

Reviewer #3: Yes

6. Review Comments to the Author

Reviewer #1: ⋅ The manuscript improved noticeably from the first version. The depth of discussion on the item selection methods seems appropriate and can be instructive for future users of the suggested methods.

⋅ It came to my notion that the simulation study was conducted using a single sample of examinees (though large). Although the study used 10 different item pools, the unstable results may be due to sampling error. If the study were replicated many times (e.g., 100), we might have observed more consistent and systematic results in Figures 3-5, and if this was the case, the consistent tendencies among the different item selection methods would become worthy of discussion.

⋅ Overall, the manuscript seems in need of minor revision to improve grammar and readability. For example,

- line 234) evaluate how the two ISRs originally proposed within the CD-CAT framework perform.

- line 287) stable results

- ‘Figure’ instead of ‘Fig’

⋅ On a minor note, the authors argued that the quadrature points do not affect measurement accuracy. However, the fineness of quadrature points does impact estimation accuracy.

Reviewer #3: I have read the revision. The authors have well addressed all my comments, and I recommend for publication.

7. PLOS authors have the option to publish the peer review history of their article (what does this mean?). If published, this will include your full peer review and any attached files.

Reviewer #1: No

Reviewer #3: No

---

## [Author Response · Author response to Decision Letter 1]

10 Dec 2019

We have included our response in the document attached as Cover Letter.

RESPONSE TO REVIEWERS: MANUSCRIPT ID PONE-D-19-15008R1

December 10, 2019

Dear Dr. Timo Gnambs:

Allow us to start by thanking you for the opportunity to revise and resubmit our manuscript “Adapting Cognitive Diagnosis Computerized Adaptive Testing Item Selection Rules to Traditional Item Response Theory” (MANUSCRIPT ID PONE-D-19-15008R1). We would also like to thank you and the reviewers for your constructive comments. In the pages that follow, we respond to the comments brought up by the reviewer.

We hope that you will find the revised version of our manuscript suitable for publication in PLOS ONE. We look forward to hearing from you regarding our submission. 

Sincerely yours,

 

REVISION COMMENTS:

REVIEWER 1:

The manuscript improved noticeably from the first version. The depth of discussion on the item selection methods seems appropriate and can be instructive for future users of the suggested methods. 

We thank you for the kind words, as well as helpful suggestions on how we can improve our manuscript.

It came to my notion that the simulation study was conducted using a single sample of examinees (though large). Although the study used 10 different item pools, the unstable results may be due to sampling error. If the study were replicated many times (e.g., 100), we might have observed more consistent and systematic results in Figures 3-5, and if this was the case, the consistent tendencies among the different item selection methods would become worthy of discussion.

While we understand the reviewer’s concern regarding the sampling error, we would like to point out that the standard error of the average estimates for the dependent variables (e.g., RMSE) can be expected to be small. We have made clearer the total number of examinees used in our study (lines 267-268), which is larger than used in most studies in this area. Also, a common practice in this area is to use a single item bank: We have used 10.

Figures 3-5 depict the average estimates across the item banks. The standard error for the average estimate of the DVs can be obtained as: SD(DV1, …, DV10) / √10, where SD is the standard deviation for the vector including the DV computed at each of the 10 item banks. We computed 95%-confidence intervals to further explore this issue. The RMSE results for the two methods with a more irregular trend (i.e., KL and DKL) are depicted here:

We can see from these plots that:

(1) When ML is used, results for the first items (1-3) can be noisier. Other than that, the confidence intervals are generally very narrow. We also checked whether each individual item bank generally followed a similar trend and we found that it was generally the case. For example, we found that RMSE values for KL and ML at item 3 were significantly greater than those at item 2 (p = 0.03, 95-CI for the different = [.008, 0.166].

(2) It should be noted that the more interesting interpretations are those made for situations in which the CAT could have ended (e.g., 10-20 items). In the figures it can be clearly seen that the standard deviation associated with the average values is close to zero, indicating that the trends and actual values computed for each of the item banks are practically identical.

From these analyses we can conclude that increasing the number of item banks would not have any effect at those points considering that the sampling error is already sufficiently small with 10 item banks. Note that although we illustrated this using the data for RMSE, results can be extrapolated to the other dependent variables. 

In the response letter we argued why the results at the very start of the CAT (1-3 items) should be interpreted with caution considering the way the CATs were started (i.e., Dodd method and the two fictitious items to compute the likelihood). Considering the information we have discussed here, we have pointed out in the manuscript that the variability across the 10 simulated banks was generally small and always negligible when the trait level estimator was EAP and/or the CAT length was equal or greater than 4 items.

We would also like to note than increasing the number of simulated items banks from 10 to 100 would imply 136 days of computation time. For each item bank we simulated 14,000 examinees (5,000 from a normal distribution and 9,000 for specific trait values) for two estimation methods, with a test length of 20 items. Considering the computation times per item reported in the manuscript –234.8 milisecond per item when all the four different item selection rules are added–, leads to a simulation that can be measured in months for this extension from 10 to 100 item banks.

Overall, the manuscript seems in need of minor revision to improve grammar and readability. For example: - line 234: evaluate how the two ISRs originally proposed within the CD-CAT framework perform. - line 287: stable results - ‘Figure’ instead of ‘Fig’

A native English speaker revised the English language of our manuscript. Track changes was used, so all the modifications can be consulted in the revised version of the manuscript. Please note that according to PLOS ONE submission guidelines, figures should be cited as “Fig 1”, “Fig 2”, etc.

On a minor note, the authors argued that the quadrature points do not affect measurement accuracy. However, the fineness of quadrature points does impact estimation accuracy. 

We agree with the reviewer’s comment. What we meant is that the impact can be expected to be negligible considering that a very large number of quadrature points (i.e., 81) was used to approximate the integrals. The default number of quadrature points in the R packages available for CAT is much smaller. The mirtCAT package uses 31 quadrature points when the number of factors is 2 (Chalmers, 2016) and the catR package uses 33 (catR; Magis & Barrada, 2017). We agree that we did not express it correctly, so we have modified that sentence in the manuscript (line 409-410). 

Work cited:

Chalmers, R. P (2016). Generating adaptive and non-adaptive test interfaces for multidimensional item response theory applications. Journal of Statistical Software, 71(5), 1-39. doi:10.18637/jss.v071.i05

Magis, D., & Barrada, J. R. (2017). Computerized adaptive testing with R: Recent updates of the package catR. Journal of Statistical Software, 76(1), 1-19. doi:10.18637/jss.v076.c01

REVIEWER 3:

I have read the revision. The authors have well addressed all my comments, and I recommend for publication.

We appreciate your comments and suggestions to improve the manuscript.

---

## [Editor Report · Decision Letter 2]

16 Dec 2019

Adapting Cognitive Diagnosis Computerized Adaptive Testing Item Selection Rules to Traditional Item Response Theory

PONE-D-19-15008R2

Dear Dr. Barrada,

We are pleased to inform you that your manuscript has been judged scientifically suitable for publication and will be formally accepted for publication once it complies with all outstanding technical requirements.

With kind regards,

Timo Gnambs

Academic Editor

PLOS ONE
---

## [Editor Report · Acceptance letter]

27 Dec 2019

PONE-D-19-15008R2 

Adapting Cognitive Diagnosis Computerized Adaptive Testing Item Selection Rules to Traditional Item Response Theory 

Dear Dr. Barrada:

I am pleased to inform you that your manuscript has been deemed suitable for publication in PLOS ONE. Congratulations! Your manuscript is now with our production department. 

With kind regards,

on behalf of

Dr. Timo Gnambs 

Academic Editor

PLOS ONE